# Identification of an insect-produced olfactory cue that primes plant defenses

Anjel M. Helms[1], Consuelo M. De Moraes [2], Armin Tröger[3], Hans T. Alborn[4], Wittko Francke[3], John F. Tooker[1] & Mark C. Mescher [2]

It is increasingly clear that plants perceive and respond to olfactory cues. Yet, knowledge about the specificity and sensitivity of such perception remains limited. We previously documented priming of anti-herbivore defenses in tall goldenrod plants (*Solidago altissima*) by volatile emissions from a specialist herbivore, the goldenrod gall fly (*Eurosta solidaginis*). Here, we explore the specific chemical cues mediating this interaction. We report that *E,S*-conophthorin, the most abundant component of the emission of male flies, elicits a priming response equivalent to that observed for the overall blend. Furthermore, while the strength of priming is dose dependent, plants respond even to very low concentrations of *E,S*-conophthorin relative to typical fly emissions. Evaluation of other blend components yields results consistent with the hypothesis that priming in this interaction is mediated by a single compound. These findings provide insights into the perceptual capabilities underlying plant defense priming in response to olfactory cues.

[1] Center for Chemical Ecology, Department of Entomology, Pennsylvania State University, University Park, PA 16802, USA. [2] Department of Environmental Systems Science, ETH Zürich, 8092 Zürich, Switzerland. [3] Institute for Organic Chemistry, University of Hamburg, 20146 Hamburg, Germany. [4] Chemistry Research Unit, USDA-ARS, Gainesville, FL 32608, USA. Correspondence and requests for materials should be addressed to M.C.M. (email: mescher@usys.ethz.ch)

Plants actively respond to the ever-changing and frequently unpredictable environments in which they live[1–3]. This responsiveness is facilitated by diverse perceptual systems that allow plants to construct information about ecologically relevant features of their environment and respond adaptively[4, 5]. Plant perceptual abilities play particularly important roles in mediating their interactions with a wide range of other organisms, including antagonists such as herbivores and pathogens[6–8]. In recent decades, the priming and induction of plant defenses in response to cues associated with herbivore and pathogen attack has become the focus of extensive research[9–13]. However, relatively little is known about the specific cues that elicit such responses in plants, or the mechanisms by which they are perceived (but see refs [14–16]).

Inducible defense responses can be described as a form of adaptive plasticity that benefits plants by allowing them to avoid unnecessary investment in defense traits in environments where antagonists are absent and to tailor their defenses to the particular antagonists encountered[9–11]. The advantages of inducible defenses may be further enhanced by defense "priming," which does not directly enhance plant resistance, but leads to more rapid and/or stronger implementation of defenses following subsequent attack[11, 12, 17, 18]. Primed defense states can be elicited by prior encounters with plant antagonists[19, 20] or by environmental cues perceived prior to attack[21, 22]. In the latter case, the reliability with which a potential cue predicts the risk of subsequent attack should be a key factor determining the efficacy of priming[23, 24].

Priming of plant defenses has previously been found to occur in response to a variety of environmental cues, including mechanical and biochemical indicators of the presence of herbivores or their eggs on plant tissues[25–30]. Recent work has also explored the priming of plant defenses via olfactory cues, focusing primarily on volatile emissions from damaged plant tissues[31–35], which have been shown to elicit defense priming in a wide range

of plant species[21, 23]. In a previous study, we demonstrated that plant defenses can also be primed by exposure to volatiles emitted directly by herbivores[36]. Specifically, we found that tall goldenrod plants (Solidago altissima) exposed to the putative male sex attractant of a specialist herbivore—the goldenrod gall fly (Eurosta solidaginis)—exhibited stronger induction of jasmonic acid (and associated defenses) in response to subsequent herbivory. Moreover, the stronger defensive responses of plants exposed to the fly emission significantly reduced herbivory relative to unexposed control plants in both laboratory and field settings. We also found that mated female E. solidaginis avoided plants previously exposed to the male emission when searching for oviposition sites. To our knowledge, this remains the only documented example of plant response to an insect-produced odor cue, although we have hypothesized that similar priming effects may occur in other co-evolved systems, where such cues reliably precede attack[36, 37].

To date, only a handful of studies have identified specific volatile cues perceived by plants, and these have focused exclusively on plant-emitted compounds[8, 15, 16, 33, 35, 38]. The primary goal of the current study was to expand our understanding of plant olfactory responses by identifying the specific cues responsible for priming of S. altissima defenses following exposure to the volatile emission of E. solidaginis. To accomplish this, we identified the volatiles emitted by male flies and assayed S. altissima defense responses to individual synthetic equivalents. A secondary goal was to investigate the sensitivity of S. altissima plants to the E. solidaginis emission by examining plant responses to different exposure levels to gain insight into the nature and capabilities of the plant's perceptual system. Documenting high sensitivity to a particular compound, for example, would suggest a finely tuned perceptual ability, perhaps mediated by a dedicated receptor system, as has been documented for plant perception of the gaseous hormone ethylene[39]. This work thus lays the

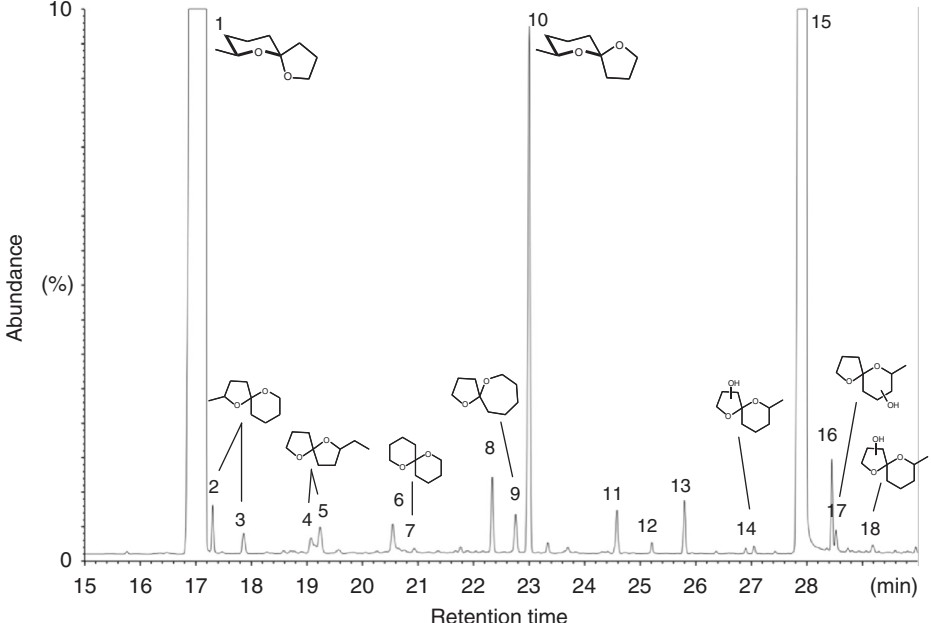

**Fig. 1** Gas chromatogram of male *E. solidaginis* volatile emission with structures of spiroacetals inserted. Structures of the conophthorins (1 and 10) show absolute configurations, whereas the others provide relative configurations. For conditions, see Methods section. List of compounds identified in the male *E. solidaginis* emission: (1) (5S,7S)-7-methyl-1,6-dioxaspiro[4.5]decane = E,S-conophthorin; (2) (2E)-2-methyl-1,6-dioxaspiro[4.5]decane; (3) (2Z)-2-methyl-1,6-dioxaspiro[4.5]decane; (4) (2Z)-2-Ethyl-1,6-dioxaspiro[4.4]nonane = Z-chalcogran; (5) (2E)-2-ethyl-1,6-dioxaspiro[4.4]nonane = E-chalcogran; (6) nonanal; (7) 1,7-dioxaspiro[5.5]undecane = olean; (8) 1-heptanol; (9) 1,6-dioxaspiro[4.6]undecane; (10) (5R,7S)-7-methyl-1,6-dioxaspiro[4.5]decane = Z,S-conophthorin; (11) 1-nonyl formate; (12) 1-octanol; (13) 1-nonyl acetate (internal standard); (14) oxygenated conophthorin I; (15) 1-nonanol; (16) (Z)-3-nonen-1-ol; (17) oxygenated conophthorin II; (18) oxygenated conophthorin III

groundwork for future studies aimed at elucidating the molecular mechanisms underlying plant perception, as well as the broader implications of plant olfaction in this system.

## Results

**E. solidaginis emissions comprise three dominant compounds.** As previously reported, male *E. solidaginis* flies emit large amounts of a volatile blend that is attractive to conspecific females and likely functions as a sex attractant[36]. Evaluation of 90 newly emerged *E. solidaginis* males revealed that the average total emission rate was $85 \pm 7\,\mu g$ over 1 day (male day equivalent, MDE; mean $\pm$ SEM), with substantial individual variation in production, ranging from $0.3\,\mu g$ to $244\,\mu g$. After collection, volatile emission samples were analyzed by gas chromatography coupled to mass spectrometry (GC/MS). Chemical components were identified using synthetic reference compounds, and their stereochemistry was assigned after enantioselective GC/MS. We found the emission blend is dominated by three compounds (listed in order of abundance): the spiroacetal (2E)-7-methyl-1,6-dioxaspiro[4.5]decane (called *E*-conophthorin); the straight-chain alcohol 1-nonanol; and the unstable *Z*-isomer of *E*-conophthorin. Together, these compounds account for >95% of the total emission. The blend also contains relatively small amounts of a few other spiroacetals, including (2E)- and (2Z)-2-methyl-1,6-dioxaspiro[4.5]decane; stereoisomers of 2-ethyl-1,6-dioxaspiro[4.4]nonane (called chalcogran); the novel 1,6-dioxaspiro[4.6]undecane; and 1,7-dioxaspiro[5.5]undecane (called olean) (Fig. 1). Details on the identification of spiroacetals and naming of specific compounds have been reviewed by Francke and Kitching[40]. The most abundant compound, *E*-conophthorin, was determined to keep (5S,7S)-configuration (hereafter called *E,S*-conophthorin) and the second most abundant spiroacetal to be its *Z,S*-diastereomer (Fig. 2). On average, male *E. solidaginis* emitted approximately $57 \pm 5\,\mu g$ *E,S*-conophthorin, $23 \pm 2\,\mu g$ 1-nonanol, $3 \pm 0.2\,\mu g$ *Z,S*-conophthorin, and only a few nanograms of the other spiroacetals over 24 h.

**A single compound primes S. altissima defenses.** To determine which compounds in the *E. solidaginis* emission prime *S. altissima* defenses, we exposed *S. altissima* plants to individual compounds of the blend and assayed insect feeding damage, as well as induction of the defense phytohormone jasmonic acid. We initially exposed plants (for 24 h) to one of five volatile treatments: (i) a crude extract of the complete *E. solidaginis* emission

blend; (ii) pure *E,S*-conophthorin; (iii) pure 1-nonanol; (iv) a racemic mixture of (2E)- and (2Z)-2-methyl-1,6-dioxaspiro[4.5]decane; (v) a dichloromethane solvent control. As noted above, *E,S*-conophthorin and 1-nonanol are by far the most abundant compounds in the blend of *E. solidaginis* volatiles, while (2E)- and (2Z)-2-methyl-1,6-dioxaspiro[4.5]decane are the second most abundant stable spiroacetals in the emission and belong to the same 1,6-dioxaspiro[4.5]decane system as *E,S*-conophthorin: only the methyl group in position 7 (adjacent to the oxygen in the six-membered ring) has been shifted to position 2 (adjacent to the oxygen in the five-membered ring).

Consistent with our previous findings[36], subsequent feeding assays revealed that larvae of the specialist beetle *Trirhabda virgata* consumed significantly less leaf tissue over 24 h compared to controls when feeding on plants exposed to the natural *E. solidaginis* emission (Fig. 3; Supplementary Table 1). Plants exposed to *E,S*-conophthorin alone exhibited a similar, and statistically significant, reduction in *T. virgata* feeding compared to unexposed control plants, indicating that exposure to this compound reduced the palatability of *S. altissima* similarly to the complete fly emission. In contrast, plants exposed to 1-nonanol experienced levels of damage similar to controls, suggesting that this compound does not influence *S. altissima* palatability to herbivores. Exposing plants to the racemic mixture of (2E)- and (2Z)-2-methyl-1,6-dioxaspiro[4.5]decane resulted in an intermediate level of damage that was not statistically distinguishable from either the *E. solidaginis* emission-exposed or control plants.

Consistent with the results of our feeding assays, we found that plants exposed to either the natural *E. solidaginis* blend or *E,S*-conophthorin exhibited an enhanced defense response, inducing significantly higher levels of jasmonic acid (JA) in response to insect feeding damage compared to controls (Fig. 4; Supplementary Table 1). In contrast, *S. altissima* exposed to 1-nonanol exhibited levels of induced JA that were similar to

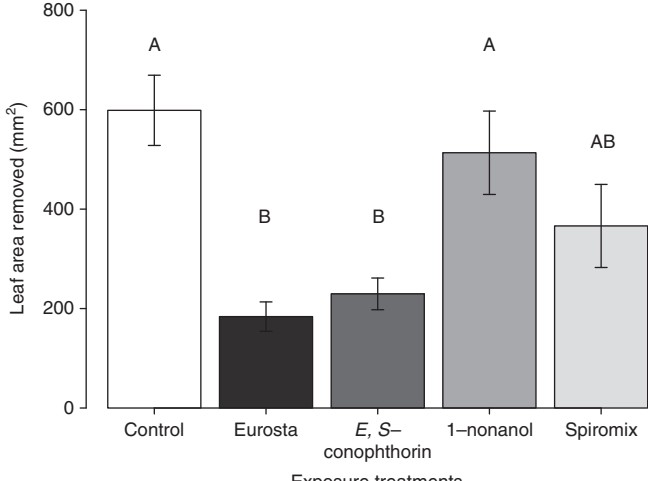

**Fig. 3** Exposure to *E,S*-conophthorin reduces feeding damage on *S. altissima*. *T. virgata* larvae consumed significantly less leaf tissue on plants exposed to the *E. solidaginis* emission blend or the most abundant compound in the blend, *E,S*-conophthorin, compared to solvent controls. *S. altissima* exposed to the second most abundant compound, 1-nonanol, received a similar amount of damage to control plants. Exposure of plants to a mixture of racemic (2E)- and (2Z)-2-methyl-1,6-dioxaspiro[4.5]decane resulted in an intermediate reduction in feeding damage (ANOVA $F_{4,25} = 9.53$, $P = 0.001$, $n = 6$). Data shown are not transformed but statistical analyses were performed on log-transformed data. *Bars marked with different letter indicates significant differences* (Tukey post hoc test, $P \leq 0.05$). *Error bars correspond to standard errors*

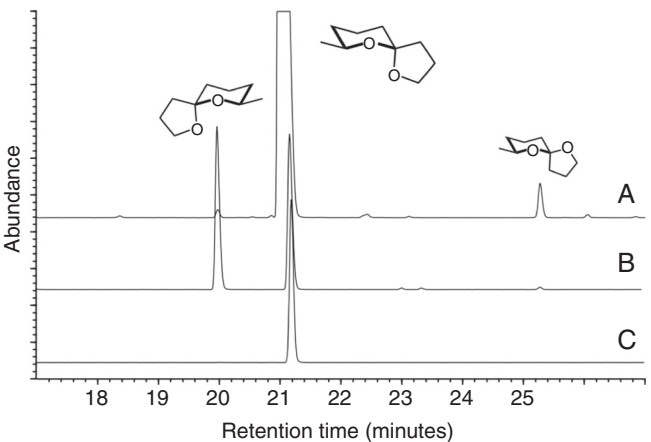

**Fig. 2** Determination of the absolute configuration of conophthorin by enantioselective gas chromatography. Track A = natural volatiles, track B = synthetic racemate of *E*-conophthorin, track C = synthetic *E,S*-conophthorin. For conditions, see Methods section

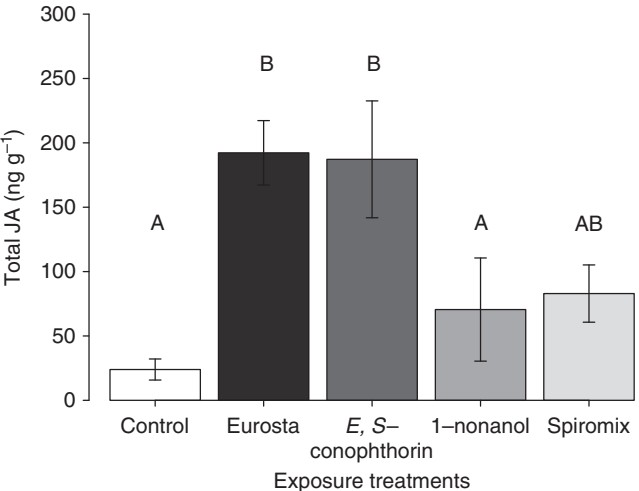

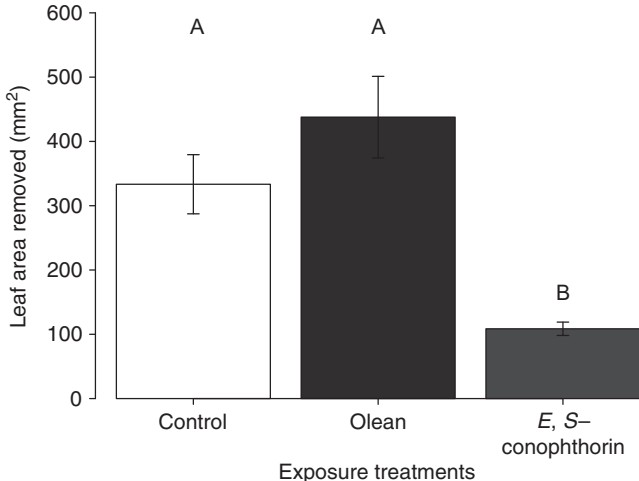

**Fig. 4** Exposure to *E,S*-conophthorin enhances jasmonic acid induction in *S. altissima*. *T. virgata* feeding damage induced significantly more JA in plants exposed to the *E. solidaginis* emission blend or the most abundant compound in the blend, *E,S*-conophthorin, compared to solvent controls. *S. altissima* exposed to the second most abundant compound, 1-nonanol, induced a similar amount of JA to control plants. Plants exposed to a racemic mix of (2E) and (2Z)-2-methyl-1,6-dioxaspiro[4.5]decane had an intermediate JA induction (ANOVA $F_{4,25} = 13.2$, $P = 0.0002$, $n = 6$). Data shown are not transformed but statistical analyses were performed on square-root-transformed data. *Bars* marked with different letter indicates significant differences (Tukey post hoc test, $P \leq 0.05$). *Error bars* correspond to standard errors

**Fig. 5** Exposure to olean does not influence feeding damage on *S. altissima*. *T. virgata* larvae consumed significantly less leaf tissue on plants exposed to *E,S*-conophthorin, compared to solvent controls or plants exposed to olean. (ANOVA $F_{2,18} = 29.4$, $P = 0.0004$, $n = 7$). Data shown are not transformed but statistical analyses were performed on log-transformed data. *Bars* marked with different letter indicate significant differences (Tukey post hoc test, $P \leq 0.05$). *Error bars* correspond to standard errors

control plants but significantly lower than the levels observed in plants exposed to the complete *E. solidaginis* emission or to *E,S*-conophthorin. Mirroring the results of our feeding assays, jasmonic acid induction in plants exposed to the racemic mixture of (2E)- and (2Z)-2-methyl-1,6-dioxaspiro[4.5]decane was not significantly different from any other treatments. These data thus reinforce our findings from the insect feeding experiment and provide further evidence that goldenrod plants respond to *E,S*-conophthorin exposure by priming their anti-herbivore defenses.

The results described above demonstrated that *E,S*-conophthorin acts as a defense priming cue for *S. altissima*, while showing no similar effect for 1-nonanol. While not significant, the intermediate effects observed for (2E)- and (2Z)-2-methyl-1,6-dioxaspiro[4.5]decane might be explained if these compounds also exhibit limited priming activity, perhaps due to their structural similarity with *E,S*-conophthorin. To further test the specificity of *S. altissima* responses to spiroacetals, we therefore conducted a complementary experiment with olean, another minor spiroacetal component of the *E. solidaginis* blend that is structurally dissimilar to *E,S*-conophthorin (comprising two six-membered rings). In this experiment, we exposed *S. altissima* plants to one of three volatile treatments: (i) the spiroacetal *E,S*-conophthorin; (ii) the spiroacetal olean; (iii) a dichloromethane solvent control. In a subsequent feeding bioassay, we found that *T. virgata* beetles consumed significantly less leaf tissue on plants exposed to *E,S*-conophthorin compared to olean-exposed and control plants (Fig. 5; ANOVA $F_{2,18} = 29.4$, $P = 0.0004$), indicating that *S. altissima* plants do not prime their defenses in response to olean exposure.

***S. altissima* responds to small amounts of *E,S*-conophthorin**. To determine the sensitivity of *S. altissima* plants to *E,S*-conophthorin, we exposed plants to different amounts of the

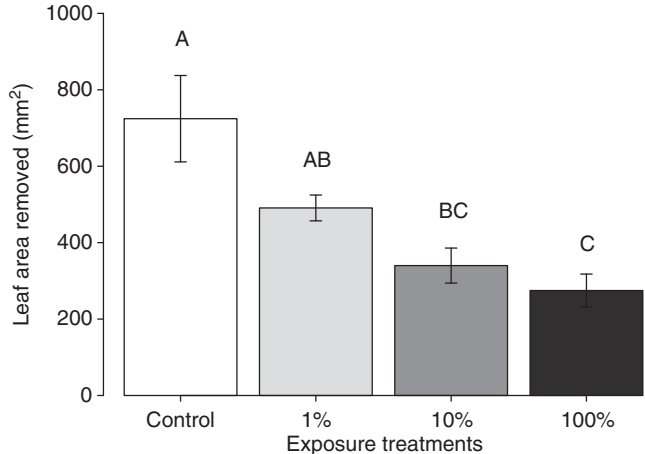

**Fig. 6** Exposure to relatively small amounts of *E,S*-conophthorin reduces feeding damage on *S. altissima*. *T. virgata* larvae consumed significantly less leaf tissue on plants exposed to a full dose of *E,S*-conophthorin, compared to unexposed control plants. The amount of damage was related to the concentration of *E,S*-conophthorin to which plants were exposed (ANOVA $F_{3,27} = 7.47$, $P = 0.003$, $n = 8$). Data shown are not transformed, but statistical analyses were performed on square-root-transformed data. *Bars* marked with different letter indicates significant differences (Tukey post hoc test, $P \leq 0.05$). *Error bars* correspond to standard errors

compound and examined the influence on *S. altissima* defenses. In a feeding bioassay, *T. virgata* beetles consumed significantly more leaf tissue on control plants compared to plants exposed to the full dose (two MDE) of *E,S*-conophthorin or a 10% dose (Fig. 6; ANOVA $F_{3,27} = 7.47$, $P = 0.003$; Supplementary Table 2). Although not statistically significant, there was also a trend toward less feeding damage on plants exposed to a 1% dose of *E,S*-conophthorin. Notably, leaf-tissue consumption by beetles exhibited a dose-dependent pattern, with plants exposed to higher doses of *E,S*-conophthorin receiving less damage. After 1 h of feeding damage by *T. virgata*, induced JA levels were significantly higher in all *E,S*-conophthorin exposure treatments compared to

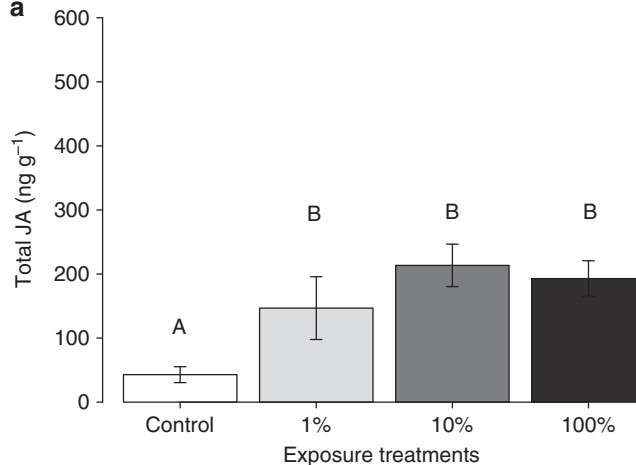

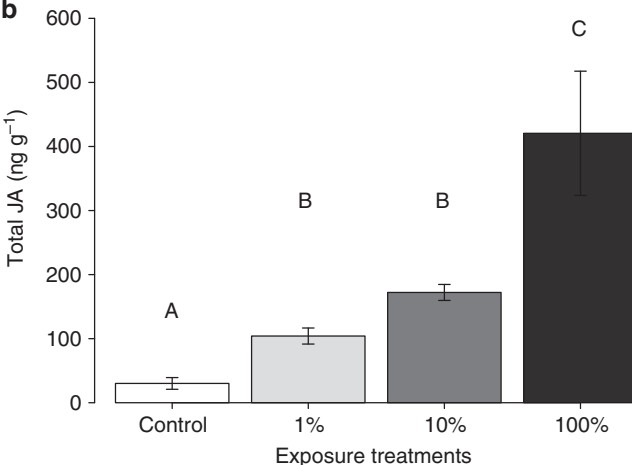

**Fig. 7** Exposure to relatively small amounts of *E,S*-conophthorin enhances jasmonic acid induction in *S. altissima*. **a** After 1 h of feeding, *T. virgata* larvae induced significantly higher levels of JA in plants exposed to *E,S*-conophthorin, compared to unexposed control plants (ANOVA $F_{3,14} = 13.09$, $P = 0.0002$, $n = 8$). **b** After 24 h of feeding by *T. virgata* larvae, *E,S*-conophthorin-exposed plants induced significantly more JA compared to controls, and induction of JA followed a dose response, with plants exposed to the full dose (100%) inducing the highest quantity of JA (ANOVA $F_{3,14} = 29.1$, $P = 0.000002$, $n = 8$). Data shown are not transformed but statistical analyses were performed on square-root-transformed data. *Bars* marked with different letter indicates significant differences (Tukey post hoc test, $P \le 0.05$). *Error bars* correspond to standard errors

control plants (Fig. 7a; ANOVA $F_{3,27} = 13.1$, $P = 0.0002$; Supplementary Table 2). After 24 h of feeding damage, plants exposed to the highest dose of *E,S*-conophthorin induced significantly more JA compared to all other treatments. Plants exposed to lower concentrations of *E,S*-conophthorin (1% or 10%) had higher levels of JA compared to control plants, but less than plants exposed to the highest dose (Fig. 7b; ANOVA $F_{3,27} = 29.1$, $P = 0.000002$; Supplementary Table 2). Similar to the insect feeding data, the JA concentrations induced after 24 h of feeding damage were also related to the exposure dose of *E,S*-conophthorin. Again, these data provide support for our findings that *S. altissima* plants exposed to small doses (1% of two MDE) of *E,S*-conophthorin exhibit enhanced anti-herbivore defenses.

## Discussion

Our data demonstrate that the most abundant component of the male *E. solidaginis* emission, *E,S*-conophthorin, primes *S.*

*altissima* anti-herbivore defenses, resulting in enhanced induction of jasmonic acid and reduced herbivore feeding damage. Specifically, we found that plants exposed to *E,S*-conophthorin or the natural emission blend induced approximately twice as much jasmonic acid compared to control plants in response to feeding damage by *T. virgata* (Fig. 4) and that beetles consumed less than half as much leaf tissue on plants exposed to the complete natural emission blend or pure *E,S*-conophthorin compared to control plants (Fig. 3). These results are consistent with our previous demonstration that the *E. solidaginis* emission primes plant defense responses in *S. altissima*[36, 37] and elucidate the specific priming cues responsible for this phenomenon.

An average male *E. solidaginis* fly typically emits about 57 μg of *E,S*-conophthorin in 24 h. This level of emission by male flies perching on *S. altissima* stems prior to mating with females would appear to provide plants with a readily detectable cue that reliably indicates increased risk of attack. Thus, the fly emission, and specifically *E,S*-conophthorin, appear to provide a salient and effective defense priming cue for the plant. However, this raises the question why male *E. solidaginis* release such large amounts of a compound that enhances plant defenses and thereby appears to reduce host plant quality. Based on the observation that the emission of *E. solidaginis* males is attractive to females, we have previously hypothesized that the emission serves as a sex attractant[36]. If so, sexual selection might explain high levels of production by males, for example, if higher emissions allow males to attract females over longer distances or if females actively utilize features of the male emission in making choices among potential mates. There is evidence that sexual selection has shaped phenotypic traits of male tephritids, including behavioral traits and pheromone production[41], and *Eurosta* males exhibit ornate wing decorations and dancing behaviors that appear to mediate courtship[42]. However, further behavioral studies are needed to definitively establish the adaptive significance of production of *E,S*-conophthorin, and the other emission components, for *E. solidaginis*.

Few studies to date have investigated the sensitivity of plants to olfactory cues with respect to either the concentration of volatiles necessary for perception or the maximum distance over which a volatile source can be perceived[43–45], although it has been suggested that plant perception of olfactory cues is predominantly a short-range phenomenon[43, 46]. Our results indicate that *S. altissima* plants respond to exposure to smaller amounts of *E,S*-conophthorin than typically emitted by males (1% of two MDE). These findings complement the results of field studies indicating that priming of defenses in neighboring plants by the *Eurosta* emission can occur over distances of at least 1 m within dense stands of *S. altissima*[47]. Thus, given the observed sensitivity of *S. altissima* to *E,S*-conophthorin, neighboring plants—which will often be ramets of the same genet as the stem on which an emitting male is perched—are also likely to exhibit priming responses. This may have ecological significance, as these adjacent stems are also likely targets for oviposition by female flies[42]. Furthermore, such responses could have important implications for the spatial distribution of *E. solidaginis* galls and other herbivorous species on *S. altissima* within a population of plants, as well as for plant-plant interactions among *S. altissima* individuals[47].

Notably, we also found that the exposure dose to *E,S*-conophthorin was positively correlated with the level of *S. altissima* defense enhancement (Figs 6, 7), indicating that there is not a simple threshold of exposure that elicits a standardized plant response. A similar pattern of increasing levels of defense with increasing volatile exposure has also been observed in lima bean plants (*Phaseolus lunatus*) exposed to vapors of methyl salicylate[48] and subsequently infested with a bacterial pathogen[44].

In contrast to E,S-conophthorin, we found little evidence that other components of the E. solidaginis blend play an important role in defense priming in S. altissima. The second most abundant compound in the emission, the straight-chain alcohol 1-nonanol, did not enhance S. altissima defenses. T. virgata beetles consumed similar amounts of leaf tissue, and elicited similar JA responses, when feeding on plants exposed to 1-nonanol and unexposed control plants (Figs. 3, 4). Thus, while 1-nonanol is emitted in relatively large amounts by E. solidaginis, it does not appear to elicit any response from S. altissima. A racemic mixture of two minor constituents of the E. solidaginis emission, (2E)- and (2Z)-2-methyl-1,6-dioxaspiro[4.5]decane, elicited an intermediate defense response that was, however, not statistically different from either control or emission-exposed plants (Figs. 3, 4). One plausible explanation for this result is that these compounds, being spiroacetals, elicit a partial or weaker response from the same perceptual mechanisms by which S. altissima recognizes and responds to E,S-conophthorin. This hypothesis is also consistent with the observation that the minor blend component olean, another spiroacetal that exhibits greater structural dissimilarity to E,S-conophthorin, did not elicit a response from S. altissima (Fig. 5).

It should also be noted that the exposure levels tested for the minor blend constituents (2E)- and (2Z)-2-methyl-1,6-dioxaspiro[4.5] were selected to match the concentration used for E,S-conophthorin and thus are far higher (by 4 orders of magnitude) than those expected to result from exposure to the fly emission itself. Consequently, it seems unlikely that either of these compounds acts as a primary priming cue for S. altissima; however, if perceivable, either could function to reinforce the information obtained via the perception of E,S-conophthorin.

Volatile spiroacetals comprise a small but relatively widespread group of natural products[40], with E-conophthorin being most commonly reported; this compound has been identified from a variety of plants[40, 49], animals[40, 50–53], and microorganisms[54–56]; and in all known cases, keeps (5S,7S)-configuration. It is interesting to note that all spiroacetals of the blend emitted by E. solidaginis, as well as 1-nonanol and (Z)-3-nonen-1-ol (Fig. 1), share a carbon skeleton comprising nine carbon atoms, a relatively widespread motif among tephritid semiochemicals (see also: El-Sayed, The Pherobase http://www.pherobase.com). Though the biosynthesis of spiroacetals is not yet fully understood[57, 58], it is likely that all volatiles identified in E. solidaginis are synthesized de novo from the acetate pool and share the same straight-chain fatty acid precursor. The oxidase system(s) accounting for the formation of spiroacetals does not seem to be highly regioselective because, as in the present case, a suite of several isomers with one or two dominating components is frequently found[59–62]. In E. solidaginis, this unspecific mid-chain oxidation is underscored by the presence of three hydroxylated conophthorins[40] among the headspace volatiles (Fig. 1: compounds 14, 17, 18). The consistent presence of this compound across different kingdoms has potential implications for interkingdom perception of volatile chemical signals. It has previously been noted that there is substantial overlap among compounds emitted by flowering plants and insects, possibly suggesting that there are few motifs or pathways available for producing volatile signals or that other evolutionary pressures lead disparate organisms to converge on similar mechanisms of semiochemical production and perception[63].

While the S. altissima-E. solidaginis system currently remains a unique example of plant response to an insect-derived odor cue, priming of plant defenses by plant-derived volatile cues appears to be widespread and has been documented for a handful of Solidago species[21], including S. altissima (Helms et al. unpublished). This raises obvious questions about whether the priming response of S. altissima to the E. solidaginis emission evolved independently or as an elaboration of a pre-existing perceptual/priming system mediating responses to plant-derived odor cues—E,S-conophthorin has not been identified among the volatile compounds emitted by S. altissima[37, 64], although it is emitted by some other plant species[40, 49]. Alternatively, perception of the E. solidaginis emission by S. altissima may have evolved independently from its ability to perceive plant-produced odors. Further studies are thus needed to elucidate the mechanisms by which plant- and insect-derived odor cues mediate defense priming in S. altissima and the similarities and differences in plant responses to these cues.

In conclusion, our findings demonstrate that the anti-herbivore defenses of S. altissima plants are primed by exposure to relatively small amounts of E,S-conophthorin, the most abundant compound in the volatile emission of E. solidaginis. As noted above, few previous studies have attempted to track plant perception of specific volatile compounds, and these have focused exclusively on plant-produced odors[8, 15, 16, 33, 35, 38]. Still fewer studies have attempted to determine how sensitive plants are to specific volatile compounds[44, 45]. Our data on the sensitivity of S. altissima to E,S-conophthorin, and the specificity of its response, thus provide novel insights into the nature of olfactory perception by S. altissima, and lay a foundation for further investigation of the mechanisms underlying S. altissima responses to the E. solidaginis emission as well their evolutionary and ecological significance.

## Methods

**The study system**. The goldenrod gall fly (E. solidaginis) induces sphere-shaped galls in stems of its host plant, tall goldenrod (S. altissima). After overwintering as larvae inside galls, E. solidaginis flies typically pupate and emerge as adults in May in the Northeastern United States. Male flies perch on the apices of goldenrod plants and attempt to attract females by rocking their bodies side to side, spreading their wings, and emitting large quantities of a putative sex attractant[36, 42, 65]. After mating, female flies seek suitable oviposition sites in the apical buds of S. altissima plants. E. solidaginis eggs hatch in 5–7 days, and larval feeding induces the formation of galls that become visible within 3 weeks[42, 65]. E. solidaginis galls reduce the reproductive fitness of S. altissima plants by diverting resources away from leaf and rhizome growth as well as inflorescence and achene production[42, 66].

**Plants and insects**. Tall goldenrod (S. altissima) plants were grown from rhizomes of the 110 and REI clone lines in insect-free, climate-controlled greenhouses, with supplemental lighting (metal halide and high-pressure sodium lights). (16 h light: 8 h dark; 24 °C: 21 °C; 60% RH). Rhizomes for these experiments were collected near State College, PA, USA, and were washed and stored at 4 °C prior to planting. Rhizomes of similar diameter were cut into 5 cm segments and planted in trays with peat-based potting soil (Pro-Mix BX; Premier Horticulture Inc., Quakertown, PA, USA). Approximately 2 weeks later, sprouted ramets were transplanted into individual pots (16 cm diameter, 16.5 cm tall) using the same potting soil, and 2 g Osmocote fertilizer (8–45–14 N–P–K, Scotts, Marysville, OH, USA) was added to each pot. S. altissima plants used in experiments were roughly 7 weeks old and 25 cm tall. Adult male E. solidaginis flies were obtained from overwintering galls collected near State College, PA, USA. Galls were placed in a climate-controlled incubator (16 h light: 8 h dark; 22 °C, 20 °C; 65% RH) for approximately 3 weeks to induce pupation and adult emergence. After emergence, male and female E. solidaginis were separated and stored at 4 °C until they were used in experiments. Adult goldenrod leaf beetles (T. virgata) were collected from fields containing S. altissima plants near State College, PA, USA. Larvae for experiments were obtained by collecting eggs from adult beetles, storing them at 4 °C for 3–5 months and then placing the eggs in moist soil in a climate-controlled greenhouse (16 h light: 8 h dark; 24 °C: 21 °C; 60% RH)[67]. Larvae typically emerged within 2–3 weeks and were fed greenhouse-grown S. altissima until they were used in experiments.

**Collection and analysis of E. solidaginis volatile emissions**. We collected male E. solidaginis emissions by aerating individual newly emerged adult male flies in 0.4-L volume glass chambers for 24 h ($n = 90$)[36, 37]. Using dynamic headspace sampling, clean, filtered air was pushed into the chambers at 0.5 L min$^{-1}$, and volatiles were trapped on two adsorbent filters containing 45 mg of Super-Q (Alltech Associates, Deerfield, IL, USA) at 0.5 L min$^{-1}$. The filters were eluted using 150 μL dichloromethane. For the exposure treatments, individual volatile samples were pooled to ensure a uniform concentration of emission. After pooling, a 150 μL aliquot of the emission solution was taken for analysis, and 5 μL of a standard containing nonyl acetate (80 ng μL$^{-1}$) and n-octane (40 ng μL$^{-1}$) was added.

Amounts of each compound in the samples were quantified using an Agilent model 7890 A gas chromatograph fitted with a flame ionization detector, using a splitless injector held at 250 °C. After sample injection, the column (HP-5, 15 m, 0.25 mm id, 0.25 μm film thickness; J&W Scientific, Folsom, CA) was maintained at 35 °C for 30 s, then ramped at 2 °C min⁻¹ to 130 °C, and ramped again at 20 °C min⁻¹ to 220 °C. In addition, a more polar VF-WAXms column was used (60 m, 0.25 mm id, 0.25 μm film; J&W Agilent Techn. Inc., USA) at 50 °C for 3 min, then ramped at 3 °C min⁻¹ to 80 °C, then ramped at 5 °C min⁻¹ to 150 °C, and ramped again at 8 °C min⁻¹ to 250 °C. Enantioselective gas chromatography was carried out with a modified cyclodextrin column (Hydrodex β-6TBDE, 25 m, 0.25 mm id; Macherey&Nagel, Düren, Germany), programed from 50 to 180 °C at 2 °C min⁻¹.

Tentative identification of target compounds was carried out by comparison of mass spectra and retention times with published data (NIST 08, NIST11, a Gothenburg Department of Chemical Ecology mass spectral library, and Wiley 7 mass spectral data base[40]), and structure assignments were unambiguously confirmed by comparison of mass spectra and GC retention times with those of authentic standards.

Racemic *E*-conophthorin, (7-methyl-1,6-dioxaspiro[4.5]decane) and its (5 S, 7 S)-stereoisomer, i.e., *E,S*-conophthorin (purity 99%), racemic mixtures of (2E)- and (2Z)-2-methyl-1,6-dioxaspiro[4.5]decane and (Z)- and (E)-2-ethyl-1,6-dioxaspiro[4.4]nonane (chalcogran), as well as a racemate of 1,6-dioxaspiro[4.6]undecane were synthesized according to Jacobson et al.[68] 1-Nonanol (purity ≥ 98.0%) and 1,7-dioxaspiro[5.5]undecane (olean) were purchased from Sigma-Aldrich (St. Louis, MO, USA).

**Emission exposure treatments**. We exposed *S. altissima* plants to the various treatments by placing individual plants in 4-L volume glass chambers that rested on Teflon bases. The glass domes and Teflon bases were cleaned with a mild detergent and water and then rinsed with acetone followed by hexanes. The stem of each plant was wrapped in clean cotton where it passed through a hole in the center of the Teflon base. Clean, filtered air was pushed into each chamber at a rate of 2.0 L min⁻¹. This airflow was used to prevent condensation or an unrealistic concentration of compounds from the exposure treatments from accumulating inside the chambers. We allowed the plants to acclimate to the glass chambers for approximately 2 h before conducting the exposure treatments.

For the individual compound exposure experiments, we exposed plants to either the natural *E. solidaginis* emission blend, pure synthetic *E,S*-conophthorin, pure synthetic 1-nonanol, synthetic (2E)-2-methyl-1,6-dioxaspiro[4.5]decane with (2Z)-2-methyl-1,6-dioxaspiro[4.5]decane, olean, or a dichloromethane solvent control ($n = 6$–7). We prepared a 0.8 μg μL⁻¹ solution of *E. solidaginis* emission crude extract in dichloromethane. We also prepared 0.6 μg μL⁻¹ solutions of *E,S*-conophthorin, 1-nonanol, (2E)-2-methyl-1,6-dioxaspiro[4.5]decane with (2Z)-2-methyl-1,6-dioxaspiro[4.5]decane, and olean in dichloromethane. We then added 45 μL of each solution to rubber septa. These doses were chosen to approximate the amount of the most abundant compound (*E,S*-conophthorin) emitted by a male fly during 6 h. We added two freshly treated rubber septa every 6 h for 24 h to allow for a more realistic exposure (2 MDE). For the dose exposure experiment, we exposed *S. altissima* plants to one of three concentrations of *E,S*-conophthorin in dichloromethane or a dichloromethane solvent control ($n = 8$). The concentrations of *E,S*-conophthorin were 0.6, 0.06, and 0.006 μg μL⁻¹. Rubber septa with 45 μL of the solution were placed into the chambers following the same procedure as the previous experiments. Again, two septa were added to each chamber at a given time to approximate a dose relative to two MDE. Sample sizes for these studies were selected based on observations from previous experiments.

**Feeding assays**. We conducted insect herbivore feeding assays as previously described[36, 37] to determine the influence of exposure to the individual *E. solidaginis* compounds on *S. altissima* anti-herbivore defenses. As discussed in our previous work, the galling habit of *E. solidaginis* larvae makes them impractical for conducting such assays; therefore, we conducted these experiments using another goldenrod specialist herbivore, the goldenrod leaf beetle (*T. virgata*) as a proxy for the fly larvae. We have previously demonstrated that exposure of *S. altissima* plants to the *E. solidaginis* emission primes anti-herbivore defenses effective against a suite of herbivores, including *T. virgata*[36].

After exposing plants to the various treatments for 24 h, we allowed *T. virgata* larvae to feed on the plants for an additional 24 h. Larvae were starved for 4 h at room temperature before the experiment. After larvae fed on the plants for 24 h, we collected the plants and scanned the leaves to measure the total leaf area consumed. The total area removed was quantified using Adobe Photoshop software.

**Quantification of jasmonic acid**. We measured levels of jasmonic acid (JA) in *S. altissima* plants as an indicator of induced anti-herbivore defenses. After plants were exposed to the various treatments for 24 h, we collected one undamaged leaf from each plant (~ 100 mg tissue). Leaves from the upper-middle section of the plant of similar size were collected. After beetles fed on the plants for 1 h or additional 24 h, we collected a recently damaged leaf from each plant, again from the upper-middle of the plant, preferably with similar damage levels, where beetles

were observed actively feeding. The tissue was flash frozen in liquid nitrogen and stored at −80 °C until analyzed. To quantify JA, we used the protocol described by Schmelz[69]. Briefly, endogenous plant hormones were extracted and derivatized to methyl esters, which were isolated using vapor-phase extraction. These compounds were then analyzed by coupled GC/CI-MS using isobutane and selected ion monitoring (SIM). We quantified amounts of jasmonic acid by adding 100 ng dihydro-JA to each sample as an internal standard. To confirm the identity of methyl jasmonate in the chromatograms, we compared the retention times and spectra of our samples with standards of the pure compound.

**Statistical analyses**. Statistical analyses were performed using the software program R (R Development Core Team (2015)). We analyzed data from the insect feeding assays using one-way ANOVA, after confirming that the data met the assumptions of normality and equal variance. Data from the individual-compound feeding experiment were log-transformed and data from the dose-response feeding experiment and all JA-data were square-root transformed to meet these assumptions. Tukey's Honest Significant Differences Test was used to conduct the post hoc multiple comparisons analysis.

**Data availability**. All relevant data are available from authors by request.

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

## Acknowledgements

Dedicated to Prof. Dr. Jerrold Meinwald on the occasion of his 90th birthday. We are grateful for the research assistance and technical support provided by E. Smyers, H. Betz and R. Sowers. This work is supported by the National Science Foundation under Grant No. DGE1255832, the Swiss National Science Foundation under Grant No. 31003A_163145, The David and Lucile Packard Foundation, and ETH Zürich. Any opinions, findings, and conclusions or recommendations expressed in this material are those of the authors and do not necessarily reflect the views of the National Science Foundation.

## Author contributions

A.M.H., C.M.D.M., M.C.M., J.F.T., and W.F. designed research; A.M.H., A.T., H.T.A., and W.F. performed research; A.M.H., A.T., and W.F. analyzed data; A.M.H., C.M.D.M., M.C.M., J.F.T., H.T.A., and W.F. wrote the paper.

## Additional information

**Competing interests:** The authors declare no competing financial interests.

