## [Peer Review File · Nature Communications]

REVIEWERS' COMMENTS:

Reviewer #1 (Remarks to the Author):

This is an exciting study. The authors provide solid chemical and behavioral evidence showing that E,S-conophthorin, a putative pheromonal compound produced by males of the goldenrod gall fly (*Eurosta solidaginis*), primes for defenses in tall goldenrod plants. The paper is nicely written and presents very convincing and unique results. I only have a few minor comments and suggestions to improve the manuscript.

In the abstract you may want to mention that the compound is produced by males only and is expected to be a sex/dispersal pheromone.

You may still want to refer to the following studies as demonstrations of the biological/ecological relevance of priming:

Van Hulten M, Pelsler M, van Loon LC, Pieterse CMJ, Ton J, 2006. Costs and benefits of priming for defense in *Arabidopsis*. *Proceedings of the National Academy of Sciences, USA* 103, 5602–7.

Ton J, D'Alessandro M, Jourdie V et al., 2007. Priming by airborne signals boosts direct and indirect resistance in maize. *The Plant Journal* 49, 16–26.

Line 95 and following: You may want to refer to the first studies that named these compounds.

Line 118: Is "system" the right term here? Would "structure" or "configuration" not be better?

Line 397: Only the highest dose represents two males, right?

Lines 407-409: Give reference.

It would be interesting to test the goldenrod responses to E,S-conophthorin in populations where the gall fly does not occur. Considering your discussion of a possible "pre-existing perceptual/priming system" (lines 277-289) it would also be interesting to study how other *Solidago* species respond to conophthorin.

The introduction and discussion could be shortened.

Reviewer #2 (Remarks to the Author):

This manuscript describes the identification of the goldenrod gall fly odorants which are, surprisingly, perceived by tall goldenrod plants. Having synthetic copies of identified volatiles, the authors then pinpointed that one of the major constituents, E,S-conophthorin, elicits a priming response of anti-herbivore defenses; plants exposed to E,S-conophthorin upregulate jasmonic acid (anti-herbivore defense plant-hormone) and are coincidentally less damaged by subsequent herbivorous attacks. Although recent works increasingly indicate that plants can perceive and respond to environmental odors, we have only limited reports that specified the key compound to elicit plant responses. The present study is particularly intriguing because the key odorant is identified from herbivorous insects as a candidate of their sex attractant pheromone. Overall, the work has in general been nicely done, and the research group has considerable experience in working with odor-based interactions among plants and associated organisms. Specific comments:

1. L.87, the sentence of "Evaluation of ninety..." is confusing. We frequently use "male day equivalent (MDE)" to indicate an emission of a pheromone from one male over 24 h. For example, "Evaluation of

ninety individuals of newly emerged *E. solidaginis* males revealed that the average total emission rate was $85 \pm 7 \mu\text{g}$ over one day (male day equivalent, MDE; mean \pm SEM), with substantial individual variation in production, ranging from 0.3 to 244 μg MDE."

2. L. 90, I agree the individual variation of volatile amounts is notable. I recommend to show this variation as a figure, for example as a histogram or a scatter plot. This may be helpful to discuss the mode of sexual selection hypothesized by the authors (L. 204), e.g. stabilizing, directional, or disruptive selection.

3. L 165, 180, 218, and others; "(two 24 h male equivalents)" -> "(two MDE)"

4. L197, "by way of comparison, many lepidopteran species...over similar time frame", I think this comparison with different taxa, different class of volatile molecules, and different emission periods, is barren and can be omitted.

5. L204, sexual selection based on pheromone emissions, i.e. correlations of pheromone emissions and mating successes, is often suggested in other tephritid flies (for example Bachmann et al. 2015 PLoS ONE doi:10.1371/journal.pone.0124250), which may support the hypothesis.

6. L262, "Z3-nonen-1-ol" -> "(*Z*)-3-nonen-1-ol"

7. L263, "Though the biosynthesis of spiroacetals is not yet fully understood..."; I think the authors should, at least, state that spiroacetals are considered to be de novo biosynthesized or derived from host plants, if possible. Tephritid flies often emit volatiles derived from their foods (e.g. Kumaran et al. 2014 J. Insect Physiol. 68: 36-43).

8. L263, "...share a carbon skeleton comprising nine carbon atoms"; C9 alcohols and their derivatives are frequently found in tephritid pheromones (El-Sayed The Pherobase <http://www.pherobase.com>), indicating these are key metabolites in tephritid and possibly supporting your idea.

9. L. 339, "We collected male *E. solidaginis*..." -> "We individually collected male *E. solidaginis*..."

10. L. 340, "small glass chamber" -> please describe its size, shape, etc.

11. L346, 375, "n-octane", "n-hexanes" -> 'n' should be in italic.

12. Figure 1 List of compounds, "2E-2-Methyl-1,6-dioxaspiro[4.5]decane" -> "(*E*)-2-Methyl-1,6-dioxaspiro[4.5]decane "

13. Figure 1 List of compounds, "2Z-2-Methyl-1,6-dioxaspiro[4.5]decane" -> "(*Z*)-2-Methyl-1,6-dioxaspiro[4.5]decane "

14. Figure 1 List of compounds, "2Z-2-Ethyl-1,6-dioxaspiro[4.4]nonane" -> "(*Z*)-2-Ethyl-1,6-dioxaspiro[4.4]nonane& #x201D;

15. Figure 1 List of compounds, "2E-2-Ethyl-1,6-dioxaspiro[4.4]nonane" -> "(*E*)-2-Ethyl-1,6-dioxaspiro[4.4]nonane& #x201D;

16. Figure 1 List of compounds, "Z3-Nonen-1-ol" -> "(*Z*)-3-Nonen-1-ol"

17. L657, 669, 677, 691, 704, "(\pm s.e.)" -> Remove.

18. Table 1, Table 2; I recommend these two tables to be provided as supplementary materials, because essential statistic scores and P-values for discussion are found in the main text.

Reviewer #3 (Remarks to the Author):

This work, done and reported to an exemplary level, shows for the first time that a herbivore pheromone component causes, at very low levels, priming in plants, *Solidago altissima*, against the same herbivore, *Eurosta solidaginis*. The compound has a sophisticated chemical structure, simplified to E,S-conophthorin and this structure, incorporating the precise stereochemistry, is essential for activity. The compound acts in priming plant defence from the gas phase and the defence priming is via the jasmonate pathway. To date we have few priming signals and these can cause unwanted plant damage. This will open up valuable new research, including of a practical nature, and will be a stimulus by providing a simple, volatile and stereochemically unique chemical priming cue to understanding more of the signal transduction pathway as, for example, we have seen for the wildfire associated germination cue karriginolide formed by plant pyrolysis. This latter work and generically

those studies on the hydroxybutenolides have been formative and pioneering but only for induction, with this work we now have a benign cue for priming, i.e. E,S-conophthorin.

Few improvements are necessary but the authors could note that the well-recognised priming agent β -aminobutyric acid (BABA) is now acknowledged as a naturally occurring agent with this effect: D. Thevenet, et al. (2016) The priming molecule β -aminobutyric acid is naturally present in plants and is induced by stress *New Phytologist* 213 (2), 552-559. Also cis-jasmone which is already known as a natural defence induction agent is also known to prime defence as a volatile cue against herbivory: S. Oluwafemi et al. (2013) Priming of production in maize of volatile organic defence compounds by the natural plant activator cis-jasmone. *PLOS ONE* 8:6:e62299.

In regard specifically to the publication of this paper: A. Tamiru et al. (2011) Maize landraces recruit egg and larval parasitoids in response to egg deposition by a herbivore. *Ecology Letters* 14: 1075-1083, demonstrates a novel insect herbivore egg elicitor of plant defence but remains chemically uncharacterised unlike the study here which, by making the chemical characterisation, thereby justifies publication to a wider readership as is recommended here.

Response to previous reviews (**our comments in bold**)

Reviewer #1

This is an exciting study. The authors provide solid chemical and behavioral evidence showing that E,S-conophthorin, a putative pheromonal compound produced by males of the goldenrod gall fly (*Eurosta solidaginis*), primes for defenses in tall goldenrod plants. The paper is nicely written and presents very convincing and unique results. I only have a few minor comments and suggestions to improve the manuscript.

In the abstract you may want to mention that the compound is produced by males only and is expected to be a sex/dispersal pheromone.

We have modified the abstract to note the focus on male emissions, but given space constraints we leave discussion of the suspected role of the emission for the fly to the main text.

You may still want to refer to the following studies as demonstrations of the biological/ecological relevance of priming: Van Hulten M, Pelsler M, van Loon LC, Pieterse CMJ, Ton J, 2006. Costs and benefits of priming for defense in Arabidopsis. Proceedings of the National Academy of Sciences, USA 103, 5602–7. Ton J, D'Alessandro M, Jourdie V et al., 2007. Priming by airborne signals boosts direct and indirect resistance in maize. The Plant Journal 49, 16–26."

We have added the Van Hulten *et al.* reference to better support our background information on plant defense priming. We have not added the Ton et al. 2007 reference due to journal limits on the number of allowable references.

Line 95 and following: You may want to refer to the first studies that named these compounds.

We have added a reference to a relevant review (Francke and Kitching 2001) and included a sentence to the text directing readers to that reference for relevant details about the identification and naming of spiroacetals.

Line 118: Is "system" the right term here? Would "structure" or "configuration" not be better?

We have revised this text for clarity, but retain "system" which is more appropriate in this context than "structure" or "configuration".

Line 397: Only the highest dose represents two males, right? 
We have revised this text for clarity.

Lines 407-409: Give reference.

We added the appropriate reference (Helms et al. 2013).

It would be interesting to test the goldenrod responses to E,S-conophthorin in populations where the gall fly does not occur. Considering your discussion of a possible "pre-existing perceptual/priming system" (lines 277-289) it would also be interesting to study how other *Solidago* species respond to conophthorin.

We are currently planning some work along these lines.

Reviewer #2

This manuscript describes the identification of the goldenrod gall fly odorants which are, surprisingly, perceived by tall goldenrod plants. Having synthetic copies of identified volatiles, the authors then pinpointed that one of the major

constituents, E,S-conophthorin, elicits a priming response of anti-herbivore defenses; plants exposed to E,S- conophthorin upregulate jasmonic acid (anti-herbivore defense plant-hormone) and are coincidentally less damaged by subsequent herbivorous attacks. Although recent works increasingly indicate that plants can perceive and respond to environmental odors, we have only limited reports that specified the key compound to elicit plant responses. The present study is particularly intriguing because the key odorant is identified from herbivorous insects as a candidate of their sex attractant pheromone. Overall, the work has in general been nicely done, and the research group has considerable experience in working with odor-based interactions among plants and associated organisms. Specific comments:

1. L.87, the sentence of "Evaluation of ninety..." is confusing. We frequently use "male day equivalent (MDE)" to indicate an emission of a pheromone from one male over 24 h. For example, "Evaluation of ninety individuals of newly emerged E. solidaginis males revealed that the average total emission rate was $85 \pm 7 \mu\text{g}$ over one day (male day equivalent, MDE; mean \pm SEM), with substantial individual variation in production, ranging from 0.3 to 244 μg MDE.

We have clarified the text and adopted the suggested use of "male day equivalent (MDE)".

2. L. 90, I agree the individual variation of volatile amounts is notable. I recommend to show this variation as a figure, for example as a histogram or a scatter plot. This may be helpful to discuss the mode of sexual selection hypothesized by the authors (L. 204), e.g. stabilizing, directional, or disruptive selection.

While the variation among individual male flies bears noting, we are limited in what we can say about its significance based on current data. For example, we don't yet know whether there is a genetic basis to the observed variation. We are currently exploring the role of the emission in the sociobiology of the fly, and hope to address this in more detail in future publications. However, we think it appropriate to keep speculation on this point to a minimum in the

current paper, which focuses on the specific cues responsible for the observed priming response.

3. L 165, 180, 218, and others; "(two 24 h male equivalents)" -> "(two MDE)

We now use MDE throughout the text.

4. L197, "by way of comparison, many lepidopteran species...over similar time frame", I think this comparison with different taxa, different class of volatile molecules, and different emission periods, is barren and can be omitted.

We omitted the comparison between *E. solidaginis* and lepidopteran pheromone production.

5. L204, sexual selection based on pheromone emissions, i.e. correlations of pheromone emissions and mating successes, is often suggested in other tephritid flies (for example Bachmann et al. 2015 PLoS ONE doi:10.1371/journal.pone.0124250), which may support the hypothesis.

We have included a reference to Bachmann et al 2015 to strengthen our discussion of sexual selection on pheromone production in tephritids.

6. L262, "*Z3-nonen-1-ol*" -> "*(Z)-3-nonen-1-ol*"

corrected

7. "L263, "Though the biosynthesis of spiroacetals is not yet fully understood..."; I think the authors should, at least, state that spiroacetals are considered to be de novo biosynthesized or derived from host plants, if possible. Tephritid flies often emit volatiles derived from their foods (e.g. Kumaran et al. 2014 J. Insect Physiol. 68: 36-43)."

We have revised the text to address this point.

8. L263, "...share a carbon skeleton comprising nine carbon atoms"; C9 alcohols and their derivatives are frequently found in tephritid pheromones (El-Sayed The Pherobase <http://www.pherobase.com>), indicating these are key metabolites in tephritid and possibly supporting your idea.

We have revised the text to address this point.

9. We individually collected male *E. solidaginis*...

We have revised the text for clarity.

10. *small glass chamber*" -> *please describe its size, shape, etc.*

We have added this information.

11. "L346, 375, "n-octane", "n-hexanes" -> 'n' should be in italic.

Corrected.

12. Figure 1 List of compounds, "2E-2-Methyl-1,6-dioxaspiro[4.5]decane" -> "(E)-2-Methyl-1,6-dioxaspiro[4.5]decane"

13. Figure 1 List of compounds, "2Z-2-Methyl-1,6-dioxaspiro[4.5]decane" -> "(Z)-2-Methyl-1,6-dioxaspiro[4.5]decane"

14. Figure 1 List of compounds, "2Z-2-Ethyl-1,6-dioxaspiro[4.4]nonane" -> "(Z)-2-Ethyl-1,6-dioxaspiro[4.4]nonane"

15. Figure 1 List of compounds, "2E-2-Ethyl-1,6-dioxaspiro[4.4]nonane" -> "(E)-2-Ethyl-1,6-dioxaspiro[4.4]nonane"

16. Figure 1 List of compounds, "Z3-Nonen-1-ol" -> "(Z)-3-Nonen-1-ol"

We revised the names as suggested throughout the text; however the “2” is needed to unambiguously assign the structure of the compounds.

17. L657, 669, 677, 691, 704, “(± s.e.)” -> Remove.

Corrected.

18. Table 1, Table 2; I recommend these two tables to be provided as supplementary materials, because essential statistic scores and P-values for discussion are found in the main text.

We have moved these tables to supplementary information, as suggested.

Reviewer #3

This work, done and reported to an exemplary level, shows for the first time that a herbivore pheromone component causes, at very low levels, priming in plants, *Solidago altissima*, against the same herbivore, *Eurosta solidaginis*. The compound has a sophisticated chemical structure, simplified to E,S-conophthorin and this structure, incorporating the precise stereochemistry, is essential for activity. The compound acts in priming plant defence from the gas phase and the defence priming is via the jasmonate pathway. To date we have few priming signals and these can cause unwanted plant damage. This will open up valuable new research, including of a practical nature, and will be a stimulus by providing a simple, volatile and stereochemically unique chemical priming cue to understanding more of the signal transduction pathway as, for example, we have seen for the wildfire associated germination cue karrikinolide formed by plant pyrolysis. This latter work and generically those

studies on the hydroxybutenolides have been formative and pioneering but only for induction, with this work we now have a benign cue for priming, i.e. E,S-conophthorin.

Few improvements are necessary but the authors could note that the well-recognised priming agent β -aminobutyric acid (BABA) is now acknowledged as a naturally occurring agent with this effect: D. Thevenet, et al. (2016) The priming

molecule β -aminobutyric acid is naturally present in plants and is induced by stress
New Phytologist 213 (2), 552-559.

We have added a reference to Theyenet et al. (2016) to our discussion of plant defense priming.

Also cis-jasmine which is already known as a natural defence induction agent is also known to prime defence as a volatile cue against herbivory: S. Oluwafemi et al. (2013) Priming of production in maize of volatile organic defence compounds by the natural plant activator cis-jasmone. PLOS ONE 8:6:e62299.

In regard specifically to the publication of this paper: A. Tamiru et al. (2011) Maize landraces recruit egg and larval parasitoids in response to egg deposition by a herbivore. Ecology Letters 14: 1075-1083, demonstrates a novel insect herbivore egg elicitor of plant defence but remains chemically uncharacterised unlike the study here which, by making the chemical characterisation, thereby justifies publication to a wider readership as is recommended here."

While these references are interesting and relevant, we have not included them due to journal limits on the total number of references allowed.